# Chemistry and Pharmacology of Delta-8-Tetrahydrocannabinol

**DOI:** 10.3390/molecules29061249

**Published:** 2024-03-11

**Authors:** Maged S. Abdel-Kader, Mohamed M. Radwan, Ahmed M. Metwaly, Ibrahim H. Eissa, Arno Hazekamp, Mahmoud A. ElSohly

**Affiliations:** 1Department of Pharmacognosy, College of Pharmacy, Prince Sattam bin Abdulaziz University, Al-Kharj 11942, Saudi Arabia; 2Pharmacognosy Department, Faculty of Pharmacy, Alexandria University, Alexandria 21215, Egypt; mradwan@olemiss.edu; 3National Center for Natural Products Research, School of Pharmacy, University of Mississippi, University, MS 38677, USA; melsohly@olemiss.edu; 4Pharmacognosy and Medicinal Plants Department, Faculty of Pharmacy (Boys), Al-Azhar University, Cairo 11884, Egypt; ametwaly@azhar.edu.eg; 5Pharmaceutical Medicinal Chemistry & Drug Design Department, Faculty of Pharmacy (Boys), Al-Azhar University, Cairo 11884, Egypt; ibrahimeissa@azhar.edu.eg; 6Hazekamp Herbal Consulting, 6420 Leiden, The Netherlands; hazekamp.hc@gmail.com; 7Department of Pharmaceutics and Drug Delivery, School of Pharmacy, University of Mississippi, University, MS 38677, USA

**Keywords:** Δ^8^-THC, isolation, chemistry, analysis, pharmacology

## Abstract

*Cannabis sativa* is one of the oldest plants utilized by humans for both economic and medical purposes. Although the use of cannabis started millennia ago in the Eastern hemisphere, its use has moved and flourished in the Western nations in more recent centuries. *C. sativa* is the source of psychoactive cannabinoids that are consumed as recreational drugs worldwide. The C21 aromatic hydrocarbons are restricted in their natural occurrence to cannabis (with a few exceptions). Delta-9-tetrahydrocannabinol (Δ^9^-THC) is the main psychoactive component in cannabis, with many pharmacological effects and various approved medical applications. However, a wide range of side effects are associated with the use of Δ^9^-THC, limiting its medical use. In 1966, another psychoactive cannabinoid, Delta-8-tetrahydrocannabinol (Δ^8^-THC) was isolated from marijuana grown in Maryland but in very low yield. Δ^8^-THC is gaining increased popularity due to its better stability and easier synthetic manufacturing procedures compared to Δ^9^-THC. The passing of the U.S. Farm Bill in 2018 led to an increase in the sale of Δ^8^-THC in the United States. The marketed products contain Δ^8^-THC from synthetic sources. In this review, methods of extraction, purification, and structure elucidation of Δ^8^-THC will be presented. The issue of whether Δ^8^-THC is a natural compound or an artifact will be discussed, and the different strategies for its chemical synthesis will be presented. Δ^8^-THC of synthetic origin is expected to contain some impurities due to residual amounts of starting materials and reagents, as well as side products of the reactions. The various methods of analysis and detection of impurities present in the marketed products will be discussed. The pharmacological effects of Δ^8^-THC, including its interaction with CB1 and CB2 cannabinoid receptors in comparison with Δ^9^-THC, will be reviewed.

## 1. Introduction

*Cannabis sativa* is one of the oldest plants used by humans either as a source of food, textile fiber, or for medical applications [1]. The medical uses of *C. sativa* started in Asia and the Middle East during the sixth century B.C., then it was introduced to the Western world in the nineteenth century [2,3]. The plant has been used for the treatment of many conditions, including inflammation, pain, depression, nausea, glaucoma, and neuralgia [4]. The medical use of cannabis did not flourish, due to variations in chemical composition and quality, until the first isolation of the main psychoactive component, Δ^9^-tetrahydrocannabinol (Δ^9^-THC) [5].

Due to the variation in the concentration of Δ^9^-THC in the many cannabis plant varieties, three phenotypes were recognized, mainly for legal classification reasons. The drug-type or phenotype I must contain a higher concentration of Δ^9^-THC than the non-psychoactive cannabidiol (CBD) [6,7]. Phenotypes II and III contain a higher percentage of CBD (more than 0.5%). Phenotype II must contain at least 0.3% of Δ^9^-THC in the female plant, while phenotype III has Δ^9^-THC less than 0.3% [6,7]. More than 125 cannabinoids—oxygen-containing C21 aromatic hydrocarbons—were identified from *C. sativa* grown in different places around the globe [8]. As an increasing number of cannabinoid related structures were synthesized, the term “Phytocannabinoids” was given to the natural cannabinoids found in the cannabis plant [9]. Based on their chemical structure, phytocannabinoids were classified into 11 categories, including the (-)-Δ^9^-*trans*-tetrahydrocannabinol (Δ^9^-THC) and (-)-Δ^8^-*trans*-tetrahydrocannabinol (Δ^8^-THC) types, among others. Other types include cannabidiol (CBD), cannabinol (CBN), cannabichromene (CBC), cannabigerol (CBG), cannabicyclol (CBL), cannabielsoin (CBE), cannabinodiol (CBND), cannabitriol (CBT), and Miscellaneous [6]. Δ^9^-THC expresses its pharmacological effects, including hypolocomotion, hypothermia, catalepsy, and analgesia via activation of the CB1 receptor [10]. Δ^9^-THC also possesses neuroprotective, antispasmodic, and anti-inflammatory actions mediated through the activation of CB2 and PPAR receptors [11]. Δ^9^-THC is reported to have anti-proliferative potential against breast cancer cells and lung metastases [12,13]. Experimentally, Δ^9^-THC can inhibit the cell growth of many tumors, such as melanoma, myeloma, hepatocellular carcinoma, and leukemia, both in vitro and in vivo [14,15]. 

In 1985, the U.S. Food and Drug Administration (FDA) approved an oral drug product containing synthetic Δ^9^-THC (Marinol^®^ capsules) to manage nausea and vomiting resulting from chemotherapy. Seven years later, it was approved by FDA as an appetite stimulant in HIV/AIDS patients [16].

Despite the therapeutic activities of Δ^9^-THC, it may also result in serious adverse effects, such as tachycardia and anxiety [17,18]. Furthermore, Δ^9^-THC may produce euphoria, cognitive impairments, and transient negative emotional states such as paranoia [19,20,21]. Reported data indicated that acute administration of Δ^9^-THC may induce transient psychosis in healthy volunteers, in addition to cognitive impairments and electroencephalography (EEG) patterns comparable to psychosis [22]. The use of Δ^9^-THC may lead to disability or loss of sense of time and distance intervals [23]. Users of Δ^9^-THC may also suffer from panic reactions, disconnected thoughts, disturbing changes in perception, feelings of depersonalization, delusions, and hallucinatory experiences [24].

Δ^8^-THC is another phytocannabinoid present in much smaller amounts compared to Δ^9^-THC [6]. Δ^8^-THC was first isolated from marijuana (dried leaves and flowers of cannabis) grown in Maryland, USA [25]. Δ^8^-THC is the positional isomer of Δ^9^-THC, and represents one of the minor groups of cannabinoids with the same chemical skeleton of tetrahydrocannabinoids. In 1975, Krejcí et al. reported on the characterization of the carboxylic acid precursor of Δ^8^-THC (Δ^8^-*trans*-tetrahydrocannabinolic acid A) (Δ^9^-THC acid A) from cannabis of Czechoslovakian origin [26]. Three members of the same skeleton were isolated from high-potency *C. sativa* grown at the University of Mississippi, USA, namely 10*α*-hydroxy-Δ^8^-tetrahydrocannabinol, 10*β*-hydroxy-Δ^8^-tetrahydrocannabinol and 10aα-hydroxy-10-oxo-Δ^8^-tetrahydrocannabinol [27,28]. In the plant, Δ^8^-THC is typically formed from nonenzymatic isomerization of Δ^9^-THC and/or cyclization of CBD [29]. Using chemical synthesis, Δ^8^-THC is much easier to produce than Δ^9^-THC [30]. Additionally, compounds of the Δ^8^-type are much more stable than their Δ^9^-counterparts. Indicative of its chemical stability, Δ^8^-THC was detected in a burial tomb in Jerusalem dating back to the fourth century B.C. [31]. Pharmacologically, it acts as a partial agonist to CB1 and CB2 cannabinoid receptors similar to Δ^9^-THC but with lower activity on both receptors [32]. The psychoactivity of Δ^8^-THC was claimed to be about 75% that of Δ^9^-THC [33]. The improved stability of Δ^8^-THC and its derivatives compared to Δ^9^-THC, coupled with the ease of its synthesis qualifies Δ^8^-THC to be a better lead compound for exploring the undiscovered aspects of cannabinoid receptors [34]. 

The availability and sale of Δ^8^-THC, obtained mainly from synthetic sources, has increased in the United States after the passing of the Farm Bill in 2018 [35]. One report indicated that Δ^8^-THC was more effective and better tolerated than Δ^9^-THC [36]. Based on the fact that Δ^8^-THC in commercial products is chemically synthesized in an unregulated market, the presence of impurities and byproducts, as well as residual solvents, is often likely. Some marketed Δ^8^-THC products were analyzed for the presence of such contaminants and several compounds such as 9*β*-hydroxy-hexahydrocannabinol (9*β*-OH-HHC) and its isomer, 9*α*-OH-HHC were identified. However, other impurities are still unknown [37].

The legal situation of Δ^8^-THC in the USA varies from one state to another [35]. The American FDA received more than 100 reports on adverse effects among recreational users between December 2020 and February 2022. Among these reported adverse effects, 55% required intervention, such as evaluation by emergency medical services or hospitalization [38]. Since these products are not approved nor evaluated for safety and/or analyzed for content, the cause of these adverse effects cannot be currently attributed to Δ^8^-THC alone. This review presents all the currently available data regarding the isolation, synthesis, analysis, and biological profile and regulation of Δ^8^-THC and its products.

## 2. Isolation of Δ^8^-THC and Its Derivatives

Δ^8^-THC was first isolated by Hively et al. in 1966 from the petroleum ether extract of the flowering tops and leaves of fresh cannabis grown in Maryland (USA) [25]. This extract was fractionated using silicic acid column chromatography eluted with benzene, and the column fractions rich in Δ^8^-THC were combined and further purified on silicic acid thin-layer chromatography (TLC) plates impregnated in silver nitrate using benzene as an eluent. The same research group also isolated Δ^8^-THC along with Δ^9^-THC from another sample of cannabis of Mexican origin and found that Δ^8^-THC constituted up to 10% of the total tetrahydrocannabinol content of the fresh marijuana samples of the above two cannabis varieties [25]. The chemical structure of Δ^8^-THC was identified by its total synthesis from cannabidiol (CBD) after treatment with *p*-toluene sulfonic acid using a previously reported method [39] and comparing the spectroscopic data of the product (IR, UV, NMR, and optical rotation) with those of Δ^9^-THC. Catalytic hydrogenation of both Δ^8^-THC and Δ^9^-THC gave two colorless resinous compounds with identical IR spectra and optical rotation indicating that both compounds have the same stereochemistry [25].

The isolated Δ^8^-THC was named Δ^6^-THC, while Δ^9^-THC was named Δ^1^-THC, based on the monoterpene numbering system that was dominant around the time (Figure 1).

In 1975, Hanuš and Krejči reported the isolation of Δ^8^-THC acid (Figure 2) from *C. sativa* of Czechoslovakian origin [26]. The dried flowering tops were extracted with petroleum ether and the extract was treated with 2% Na_2_CO_3_ solution followed by acidification with cold 5% H_2_SO_4_, then extracted with ether to produce an acid fraction containing cannabinoid acids. This fraction was purified over a silica gel column eluted with chloroform to yield a cannabinoid acids mixture, which was esterified by diazomethane and further purified on a silica gel column using *n*-hexane: ether (2:1) as a mobile phase to afford Δ^8^-THC acid methyl ester. Identification of this methyl ester was performed by IR, UV, and NMR spectroscopic analysis, as well as EI-Mass spectrometry [26].

From high potency *C. sativa* (≥20% *w*/*w* Δ^9^-THC) grown at the University of Mississippi, USA, three minor hydroxylated derivatives of Δ^8^-THC were isolated in 2015, and identified as 10aα-OH-10-oxo-Δ^8^-THC, 10α-OH-Δ^8^-THC and 10*β*-OH-Δ^8^-THC (Figure 2). The dried buds were sequentially extracted with hexanes, CH_2_Cl_2_, EtOAc, EtOH, EtOH/H_2_O, and H_2_O. The hexanes extract was fractionated on successive silica gel columns followed by purification on semi-preparative reversed-phase HPLC using ACN: H_2_O (75:25) to afford 10aα-OH-10-oxo-Δ^8^-THC, along with other minor oxygenated cannabinoid derivatives [28]. The CH_2_Cl_2_ and EtOAc extracts were combined and subjected to a silica gel Vacuum Liquid Chromatography column (VLC) on silica gel eluted with a gradient mixture of EtOAc/hexanes to give nine fractions (A-I). Fraction D was chromatographed on a silica gel column and then further purified on a flash C18 column using 60% aqueous MeOH as a mobile phase. Two cannabinoids, 10α-OH-Δ^8^-THC, and 10*β*-OH-Δ^8^-THC, were finally isolated using a silica HPLC column, isocratically eluted with 10% EtOAc/*n*-hexane [27].

## 3. Is Δ^8^-THC a Natural Secondary Metabolite or an Artifact?

Upon its initial isolation in 1966, Hively et al. claimed that Δ^8^-THC is a natural metabolite and not an artifact produced during the extraction and isolation procedures. The researchers performed two experiments to support their assumption: first, they subjected pure Δ^9^-THC to chromatography using the same methods described for marijuana extract. They discovered that no Δ^8^-THC was formed [25]. Second, they extracted 2-year-old Spanish marijuana and 3-year-old Mexican marijuana with petroleum ether, and only Δ^9^-THC was detected and no Δ^8^-THC, based on TLC analysis [25]. Later, Pars and Razadan (1971) studied the stability of synthetic Δ^8^-THC and Δ^9^-THC at 80˚C. Δ^8^-THC showed no change in the content, while Δ^9^-THC was totally decomposed to CBN only and no Δ^8^-THC was observed as indicated by GC analysis [40].

Turner et al. (1973) reported a 2-year stability study of the four cannabinoids, Δ^9^-THC, Δ^9^-THC acid A, CBD, and CBN in dried cannabis plant samples stored at five different temperatures (−18 °C, 4˚C, 22 °C, 37 °C, and 50 °C). GLC was used to analyze the content of each cannabinoid in all samples every 10 weeks for 104 weeks. At the end of the study, it was concluded that the concentration of Δ^9^-THC, and Δ^9^-THC acid A was decreased as the storage temperature increased, while the CBN content increased with increasing temperature, except that it decomposed at 80 °C after 50 weeks. CBD content did not change at any of the temperatures. Also, no change in the Δ^8^-THC was observed in this study [41]. In the same year, the same research group reported that chloroform is an efficient extraction solvent for *C. sativa* compared to seven other solvents (benzene, pentane, hexane, petroleum ether, ethanol, acetone, and ether), and the cannabinoids present in this extract (Δ^9^-THC, CBD, CBC, and CBN) were stable for 144 hrs based on GC analysis (before and after silylation) [42]. Δ^8^-THC was detected in trace amounts but when the GC column contains a free hydroxyl group or the extraction solvent contains H_2_O, the content of Δ^8^-THC increased. The researchers postulated that Δ^8^-THC is an artifact formed during the extraction procedure or GC analysis.

Based on the above experiments, further research is needed to answer the question, “Is Δ^8^-THC a natural (biosynthetically produced) cannabinoid or an artifact?”.

## 4. Synthesis of Δ^8^-THC

Since the concentration of the naturally occurring Δ^8^-THC in the cannabis plant is exceedingly low, its extraction holds little economic viability due to the substantial associated expenses [43,44]. Consequently, almost all the Δ^8^-THC on the market today is synthetically produced, predominantly from the chemical conversion of cannabidiol (CBD) [45]. CBD can be readily transformed into Δ^8^-THC through acid-catalyzed cyclization reaction [46,47], as shown in Figure 3. The use of common household chemicals to obtain the same chemical transformation is readily accessible, and numerous internet sources provide comprehensive step-by-step descriptions of this conversion [36].

## 5. Analysis of Δ^8^-THC

The small difference in the chemical structure of the two isomers of tetrahydrocannabinol has a noticeable effect on the three-dimensional shape of the molecule, which may account for the differences in the pharmacological and psychological effects described [48]. Δ^8^-THC shares the same mechanism of action as Δ^9^-THC as a partial agonist of the cannabinoid CB1 receptor, but with less potency [33,48,49]. The increasing availability of products containing Δ^8^-THC and its isomers, which are erroneously classified as hemp, emphasize the need for precise and accurate analytical methods to evaluate its existence and concentration in various matrices.

### 5.1. Analysis of Δ^8^-THC in Cannabis Biomass and Cannabis-Derived Products

Liquid chromatography coupled with photodiode array detection (HPLC-PDA) is commonly used for the analysis of Δ^8^-THC in samples from herbal cannabis and/or cannabis-derived products (extracts, oils, concentrates, and hash). In contrast, liquid chromatography coupled to mass spectrometry (LC-MS) is the preferred method for the determination of Δ^8^-THC and its metabolites in biological fluids. Additional analytical techniques, including gas chromatography-mass spectrometry (GC-MS), gas chromatography with flame ionization detector (GC/FID), and quantitative nuclear magnetic resonance (QNMR) were also reported for Δ^8^-THC analysis.

HPLC-PDA methods are very often utilized in association with the quantification of Δ^8^-THC in complex extracts from *C. Sativa*. In 1979, Masoud et al. reported the very first method for the analysis of Δ^8^-THC in Mexican cannabis extracts using HPLC with electrochemical detection [50]. More recently, Correia et al. (2023) developed and validated an HPLC-PDA method for the quantification of Δ^8^-THC, in addition to Δ^9^-THC, CBD, CBN, Δ^9^-THCA, and CBDA in cannabis products, following an ultrasound-assisted solid–liquid extraction protocol [51].

Recently, an HPLC method was validated for the analysis of Δ^8^-THC and other 14 cannabinoids in eight cannabis plant material samples of four chemovars (high THC, high CBD, THC/CBD, and high CBG) and two e-cigarettes. Δ^8^-THC content (%*w*/*w*) was in the range of 0.036–0.60 in high THC, high CBD, THC/CBD chemovars, while it was not detected in both CBG chemovar and cannabis e-cigarettes [52].

An HPLC-PDA method was developed by Duffy et al. in response to the e-cigarette or Vaping Product Use-Associated Lung Injury (EVALI) outbreak in New York State. The analyzed samples from cannabis vaporizer cartridges contained abnormally high amounts of Δ^8^-THC, whereas medical marijuana products typically do not. Δ^8^-THC amounts ranged from 9.6 to 98% *w*/*w* in the analyzed vaping products. Moreover, untargeted analysis by GC-MS and LC-HRMS was performed for the determination of diluents and other cartridge components in order to further understand the occurrence of EVALI [53].

Using standards in the validation phase, Δ^8^-THC could be baseline separated from ten other cannabinoids using acetonitrile (with 0.05% formic acid) and water (with 0.05% formic acid) as a mobile phase in a gradient mode [54]. This study used an Ultrahigh-Performance Liquid Chromatograph coupled with a Photodiode Array and single quadrupole Mass Spectrometry detectors (UPLC-PDA-MS) for its analysis in different *C. sativa* samples including leaves, flower buds, and hashish. However, only a negligible amount of Δ^8^-THC was found in one hashish sample [54].

Different analytical techniques have been employed for the evaluation of cannabinoid content, including Δ^8^-THC in hemp seed oil and hemp distillate. In 2019, Križman et al. developed an HPLC-PDA method, applying isocratic elution at 37 °C for the analysis of hemp plant samples. However, only one plant sample analyzed contained a detectable amount of Δ^8^-THC [55]. Based on the author’s claim, the use of 275 nm detection wavelength gave a significantly better sensitivity (signal-to-noise ratio) than the conventional 228 nm wavelength commonly used for cannabinoid analysis. In the total of four hemp samples, Δ^8^-THC was detected and quantified in only one sample at a concentration of 0.06% *w*/*w* [55].

A UPLC-PDA method was developed and validated by Song et al. (2022) for the determination of 16 cannabinoids, including Δ^8^-THC in different hemp-derived concentrates. The mobile phase composition was optimized by studying different percentages of acetonitrile and formic acid concentrations in water. The best resolution between Δ^9^-THC and Δ^8^-THC peaks was achieved using 0.028% formic acid and 73% acetonitrile [56]. Method specificity was validated by quadrupole time-of-flight (Q-TOF) mass spectrometry. Only two out of nine hemp concentrates analyzed contained Δ^8^-THC, namely one type of Δ^8^-THC distillate and one type of Δ^8^-THC hemp shatter, where Δ^8^-THC contents were 70.96% and 72.32% (*w*/*w*), respectively [56].

Generally, LC-MS/MS techniques are used for analyzing Δ^8^-THC in complex cannabis matrices, such as blood samples or food products. The technique cannot distinguish Δ^8^-THC from interfering cannabinoids such as Δ^9^-THC, CBD, CBC, and CBL, based on the analyte’s mass-to-charge ratio (*m*/*z*), as they exhibit nearly identical *m*/*z* values and share matching fragmentation spectra [57]. As a result, for accurate analysis of Δ^8^-THC, a full chromatographic separation has to be achieved.

Christinat et al. (2020) developed and validated an LC-MS/MS method for the quantification of 15 phytocannabinoids, including Δ^8^-THC in different food products. The mobile phase consisted of 0.1% formic acid solution in water (solvent A) and acetonitrile (solvent B) in a gradient elution mode. In 14 out of 23 hemp-based food products, Δ^8^-THC amounts were <1 mg/kg, while one CBD oil sample contained as much as 4.96 mg/kg [58].

An ESI-LC/MS method was reported for the metabolic profiling of phytocannabinoids in cannabis, where cannabinoid concentration from ethanolic extraction was determined using deuterated cannabinoids (Δ^8^-THC-d_9_, Δ^9^-THC-d_9_, CBD-d_3_, and CBN-d_3_) as internal standards. The identified Δ^8^-THC amount was below the limit of quantification of the reported method (1.25 ng/mL) [59]. 

In 2012, Trofin et al. developed a GC-MS method for the determination of cannabinoids with baseline separation of Δ^8^-THC and Δ^9^-THC, in different types of herbal cannabis. The analysis was performed using a fused silica capillary column and a temperature program from 150 °C to 280 °C [60].

### 5.2. Analysis of Δ^8^-THC, Impurities, and Possible Contaminants in Commercial Consumer Products

In the United States, commercial products containing Δ^8^-THC in high concentrations are gaining popularity, since the 2018 Farm Bill made tremendous changes to the regulation of hemp products [61]. A wide variety of Δ^8^-THC products are being sold on the shelves of shops, gas stations, or from online sources.

These products are manufactured from CBD derived from CBD-rich hemp flowers, mainly based on the acid cyclization of cannabidiol (CBD) as the reactant (Figure 4). The cyclization reaction categorizes the resulting Δ^8^-THC by some scientists as a synthetic cannabinoid, thereby designating it as a controlled substance [46]. However, the discussion on the legality of Δ^8^-THC is ongoing for now. According to the stated arguments, Δ^8^-THC in commercial products is typically a synthetic product; consequently, there are valid concerns regarding the presence of impurities in these products with unknown effects on the human body. A recent FDA consumer warning update (2022) addressed several concerns with the chemical manufacturing process mentioned above [38]. Some manufacturers may use potentially unsafe household chemicals to make Δ^8^-THC through this chemical synthesis process. Additional undisclosed chemicals may be used to change the color of the final product or to increase shelf life. The final Δ^8^-THC product may have potentially harmful by-products (contaminants), and there is uncertainty with respect to other potential contaminants that may be present or produced depending on the composition of the raw starting materials. When consumed or inhaled, contaminants from the synthetic starting materials, as well as byproducts of the synthetic products of Δ^8^-THC, can be harmful. 

Δ^8^-THC marketed products were found to contain unnatural cannabinoids suggesting these were of synthetic origin [37].

As a result of the synthetic production methods and the lack of regulatory requirements, hemp-derived products containing Δ^8^-THC and other isomers are at greater risk of contamination and adulteration than herbal cannabis products sourced from licensed cannabis producers for medical or adult use. By the year 2020, the Interim Final Rule (2020 IFR) issued by the Drug Enforcement Administration (DEA) confirmed that hemp-derived THC products are not legalized by the 2018 Farm Bill [62]. As a result, the increased proportion of Δ^8^-THC and impurities present in various non-traditional marijuana-containing products, such as vape cartridges, extracts, and edibles, has become an important analytical challenge.

In order to address this emerging class of Δ^8^-THC containing consumer products, different analytical techniques have been reported in the literature. Radwan et al. (2023) isolated and characterized impurities found in synthesized Δ^8^-THC raw material in the USA using GC-MS, one Dimension and two Dimension Nuclear Magnetic Resonance spectroscopic (1D and 2D NMR) analysis, as well as High-Resolution Electron Spray Ionization Mass Spectrometry (HR-ESI-MS) [37]. The identified compounds included olivetol, Δ^4,8^-iso-tetrahydrocannabinol (Δ^4,8^-iso-THC), iso-tetrahydrocannabifuran (iso-THCBF), cannabidiol (CBD), Δ^8^-iso-tetrahydrocannabinol (Δ^8^-iso-THC), Δ^8^-cis-iso-tetrahydrocannabinol (Δ^8^-cis-iso-THC), 4,8-epoxy-iso-tetrahydrocannabinol (4,8-epoxy-iso-THC), Δ^9^-THC, 8-hydroxy-iso-THC (8-OH-iso-THC), 9α-hydroxyhexahydrocannabinol (9α-OH-HHC), and 9β-hydroxyhexahydrocannabinol (9β-OH-HHC) (Figure 5).

The presence of the precursor molecule olivetol suggests a fully synthetic Δ^8^-THC in these products. This hypothesis is supported by some studies that reported the synthesis of tetrahydrocannabinols from olivetol as a precursor [63,64]. The presence of olivetol was also reported by Meehan-Atrash and Rahman (2021) as a contaminant in 27 Δ^8^-THC products [65]. The study found that none of the analyzed products had accurate labeling of Δ^8^-THC content (within a range of ±20% of the concentration on the label). Moreover, all the products contained reaction by-products in addition to olivetol, including Δ^4,8^-iso-THC, 9-ethoxyhexahydrocannabinol (9-EtO-HHC), Δ^9^-THC, and a previously unidentified cannabinoid (iso-THCBF, Figure 5) in addition to heavy metals [65]. Quantitative NMR and GC-MS methodology were also used to analyze the major components of 27 e-cigarette products advertising Δ^8^-THC. All products were found to contain at least some undisclosed diluents and/or reaction side-products from the conversion of CBD to Δ^8^-THC. Among others, triethyl citrate (diluent; present in 7 out of 27 samples), olivetol (likely the starting material of the chemical synthesis in 22/27 samples), and Δ^8^-THC -iso-THC (a byproduct of the acid-catalyzed CBD cyclization; 27/27 samples) were identified (Figure 5). 

In addition, the previously unknown cannabinoid (5aR,9aS)-5a-isopropyl-8-methyl-3-pentyl-5a,6,7,9a-tetrahydrodibenzo[b,d]furan-1-ol or iso-THCBF was found (Figure 5). It was present in nearly all products tested, but was not quantifiable in products containing the diluent triethyl citrate due to spectral overlap. Analysis of heavy metals by ICP-MS confirmed detectable levels of magnesium, chromium, nickel, copper, zinc, mercury, lead, and silicon [65]. Moreover, QNMR indicated that Δ^8^-THC levels can vary as much as 40% from the labeled value, suggestive of poor testing capabilities or perhaps even falsified results. For one of the brands tested, the average of the sums of Δ^8^-THC and Δ^8^-iso-THC for each product was not significantly different from the average reported Δ^8^-THC content, suggesting the analysis method (HPLC-UV as stated in the certificate of analysis) cannot discriminate between the two compounds [65].

Ray et al. (2022) used ^1^H-NMR, HPLC-UV, and HPLC-MS to determine unknown impurities in Δ^8^-THC consumer products purchased online or from local retailers from the USA [66]. Ten Δ^8^-THC products, including distillates and vaporizers, were analyzed during the study. The results indicated that the tested samples contain a range of impurities in concentrations far beyond what is declared on certificates of analysis for these products. Impurities were identified as compounds resulting from low-quality CBD (extracted from hemp used as starting material), as well as known side reaction products from the cyclization reaction used to convert CBD into Δ^8^-THC. The identified compounds included cannabidivarin (CBDV) and cannabidihexol (CBDH). The study concluded that the problem with these products is threefold: impure CBD starting material, poor post-reaction purification, and inadequate separation of closely related cannabinoids during laboratory analysis of the final product [66].

In one study, in addition to Δ^8^-THC, the cannabinoids Δ^9^-THC, CBD, 9(S/R)-Δ^6a,10a^-THC, and (6aR,9R)-Δ^10^-THC were identified in an e-cigarette cartridge. All of these are isomers of Δ^9^-THC. The identification was based on retention time and fragmentation patterns compared with reference materials. Moreover, some unknown components were also detected within the vape oil of the same e-cigarette [44].

The main cause of the presence of THC isomers is probably due to exposure of cannabis source materials, such as cannabis concentrates or converted hemp materials, to chemical and thermal treatments during the manufacturing process. Using GC-MS and HPLC-PDA analyses, some isomers have been identified in many of the vaping liquids and distillates including Δ^8^-THC, Δ^9^-THC, 9R-Δ^6a,10a^-THC, 9S-Δ^6a,10a^-THC, 6aR,9R-Δ^10^-THC, 6aR,9S-Δ^10^-THC, Δ^6a,10a^-THC, exo-THC, Δ^6a,10a^-tetrahydrocannabivarin (Δ^6a,10^-THCV), THCA, Δ^9^-THCV, Δ^8^-THCV, CBD, CBDV, CBN, CBG, and CBC [67]. Significant amounts of Δ^8^-THC were found in the analyzed vaping liquids and distillates, ranging from 0.4 to 79% *w*/*w*, while Δ^9^-THC amounts ranged from 1.4 to 93.1% *w*/*w* as determined by GC-MS analysis. In addition, low amounts of the Δ^8^-THCV, Δ^9^-THCV, and CBDV were determined. The chemical structures of THC isomers are depicted in Figure 6.

### 5.3. Analysis of Δ^8^-THC and Δ^9^-THC Metabolites in Different Biological Matrices

In human liver microsomes, Δ^8^-THC is enzymatically metabolized similar to Δ^9^-THC. In brief, Δ^8^-THC is oxidized to the pharmacologically active, 11-hydroxy-Δ^8^-tetrahydrocannabinol (11-OH-Δ^8^-THC) [68,69], and then further to the non-psychoactive 9-carboxy-11-nor-Δ^8^-tetrahydrocannabinol (Δ^8^-THC-COOH), which is finally conjugated to form its corresponding glucuronide before being excreted, mainly in urine [70,71,72]. Figure 7 shows the metabolic scheme for Δ^8^-THC and Δ^9^-THC.

The terminal elimination half-life of Δ^8^-THC has not yet been reported in humans. The terminal half-life of Δ^9^-THC is up to 4 days in chronic users [70]. This long half-life is attributed to the slow release of the highly lipophilic Δ^9^-THC from fat tissue to plasma. Given the similar distribution to fat tissue, it is expected that Δ^8^-THC will have a similarly long terminal half-life, which means that Δ^8^-THC metabolites may be detected for days or even weeks after the last consumption [70]. Although human data on the metabolism of Δ^8^-THC is limited, some information has been learned from cases involving accidental overdose [73].

At the Karolinska University Laboratory in Sweden, and during routine LC-MS/MS analysis of the human Δ^9^-THC metabolite Δ^9^-THC-COOH in urine, an unidentified peak was detected with a slightly different retention time than Δ^9^-THC-COOH. Using an optimized LC-HRMS method with better separation between the two peaks, the unknown peak was confirmed to be Δ^8^-THC COOH [74].

Pharmacokinetic studies were conducted using the LC-MS method for the determination of Δ^8^-THC and its metabolite Δ^8^-THC-COOH in guinea pig plasma. A liquid–liquid extraction protocol was developed to extract the target analytes from guinea pig plasma using acetonitrile: ethyl acetate (50:50, *v*/*v*) [75].

Rzeppa et al. (2021) used HPLC-MS/MS for the analysis of Δ^8^-THC-COOH as a less common metabolite compared to Δ^9^-THC-COOH. The authors also carried out a GC-MS/MS for simultaneous analysis of Δ^8^-THC-COOH. The content of Δ^8^-THC-COOH in doping samples ranged from 0.05 to 2.83% [43]. These findings are in line with those made by in vitro experiments using human liver microsomes [70].

LC-QTOF/MS was used for the analysis of Δ^9^-THC and Δ^8^-THC in the serum and urine of a 2-year-old girl accidentally intoxicated with Δ^8^-THC-infused gummies [76]. The analysis confirmed the presence of Δ^8^-THC and the absence of Δ^9^-THC in the gummy. Plasma concentrations of Δ^8^-THC and Δ^8^-THC-COOH were 107.6 ng/mL and 746.5 ng/mL, respectively, while urine analysis exhibited 1550 ng/mL of Δ^8^-THC-COOH. The study revealed that the symptoms observed in individuals exposed to Δ^8^-THC closely resemble the established effects of Δ^9^-THC. These resemblances pose challenges for clinicians attempting to pinpoint the exact substance that has been ingested.

Another LC-MS/MS method was developed for the analysis of Δ^8^-THC and Δ^9^-THC and their metabolites in blood. Analyte extraction was carried out using 10 mM phosphate-buffered saline, 1% phosphoric acid, and mixed solvent (80:10:10 hexane/ethyl acetate/methyl-tert-butyl-ether) [44].

Analysis of 1504 urine specimens with a positive immunoassay Δ^9^-THC initial test using a liquid chromatography-tandem mass spectrometry (LC-MS-MS) showed the presence of Δ^8^-THC-COOH in 378 samples (15 ng/mL, cutoff), compared to 1144 specimens containing Δ^9^-THC-COOH [77].

Isomeric separation was achieved between Δ^8^-THC-COOH and Δ^9^-THC-COOH by using an automated online μSPE-LC-MS/MS method. Δ^8^-THC-COOH were detected in 54 out of 78 urine samples. An Acquity UPLC HSS T3 (2.1 × 100 mm, 1.8 µm, Waters, Milford, MA, USA) column was used for chromatographic separation. The mobile phase consisted of 0.05% acetic acid in water and methanol with gradient elution [78].

For many years, oral fluid has typically been used for drug testing. Different extraction methods were employed for Δ^8^-THC analysis in oral fluid. In order to avoid strong acidic conditions preventing CBD conversion to Δ^8^-THC, LC-MS/MS methods using solid phase extraction and solid phase microextraction methods were optimized for the measurement of cannabinoids including Δ^8^-THC in oral fluid (Table 1) [79,80,81].

An HPLC-MS/MS method was developed for the analysis of Δ^8^-THC, Δ^9^-THC, CBD, and 10 additional cannabinoids and their metabolites in oral fluid (including 11-OH-Δ^9^-THC, Δ^9^-THC-COOH, THCV, CBDV, cannabidiorcol (CBD-C_1_), CBC, CBN, and CBG [79]. Baseline separation between Δ^8^-THC and Δ^9^-THC peaks was achieved using a CORTECS^®^ Cl8 analytical column at 26 °C under a gradient elution of the mobile phase (A) 0.1% formic acid in water/acetonitrile (95:5, *v*/*v*) and (B) 0.1% formic acid in acetonitrile. In 11 out of the 200 oral fluid samples, Δ^8^-THC was detected at concentrations ranging from 0.2 to 339.5 ng/mL. In the same study, two multiple reaction monitoring transitions (MRMs) were monitored for Δ^8^-THC; a quantifier; 315.1 → 123.0 (*m*/*z*) and a qualifier 315.1 → 135.1 (*m*/*z*), and only one MRM transition was monitored for the deuterated internal standard of Δ^8^-THC (Δ^8^-THC-d_9_); 324.21 → 123.l (*m*/*z*) for the analysis of Δ^8^-THC. The results showed that only one sample contained Δ^8^-THC and its concentration was as high as 925.7 ng/mL, suggesting possible mucosa saturation [79].

A 2D Ultra-Fast Liquid Chromatography tandem mass spectrometry (UFLC-MS-MS) method was developed and validated to separate Δ^8^-THC and Δ^9^-THC isomers and their metabolites in human blood. A liquid–liquid extraction protocol was used for extracting the target cannabinoids from whole blood. Complete chromatographic separation of Δ^8^-THC-COOH from Δ^9^-THC-COOH was achieved over a run time of 10 min. The authors claimed that “this is the first report of a method that successfully quantitates these primary cannabinoids in blood specimens where significant concentrations of both Δ^8^-THC and Δ^9^-THC isomers are present” [44].

### 5.4. Overestimation of Δ^9^-THC-COOH Levels: A Special Concern

Due to structural similarities, the existence of Δ^8^-THC-COOH in a urine specimen could potentially negatively interfere with the testing process for Δ^9^-THC-COOH. Consequently, LC-MS-based analytical methods may experience identification problems with the two isomers, unless they are chromatographically well separated. This is because they have the same molecular mass, and Δ^8^-THC-COOH also showed the same most abundant MS transitions, thereby often meeting the MS acceptance requirements for a positive identification as Δ^9^-THC-COOH. There have been several studies demonstrating interfering peaks during the detection and/or quantitation of Δ^9^-THC-COOH by LC-MS/MS or GC-MS [79].

Hadener et al. (2017) expressed special concern about the potential overestimation of Δ^9^-THC-COOH blood levels due to the existence of its Δ^8^ isomer, using LC-ESI-MS/MS. This could contribute to inconsistencies in Δ^9^-THC-COOH concentrations reported by different laboratories [82].

A drug screening by immunoassay is typically the first step in drug testing for both forensic and clinical samples. The cross-reactivity immunoassay of Δ^8^-THC-COOH in the EMIT II phase was assessed at a cutoff of 20 ng/mL. The authors carried out a confirmatory GC-MS method to prevent false positive results of overestimation of Δ^9^-THC-COOH. Moreover, the reported GC-MS method succeeded in the separation and quantification of both Δ^9^-THC-COOH and Δ^8^-THC-COOH as TMS derivatives using relative retention time compared to Δ^9^-THC-COOH-d_9_ as the internal standard [83].

A baseline separation of Δ^8^-THC-COOH and Δ^9^-THC-COOH peaks has been achieved using GC-MS analysis. Although Δ^8^-THC-COOH and Δ^9^-THC-COOH show common fragmentation ions, they were eluted 0.05 min apart, resulting in baseline resolution. The developed method was then applied for the analysis of Δ^8^-THC-COOH and Δ^9^-THC-COOH in postmortem urine samples. A total of 26 out of 194 postmortem urine samples were positive for Δ^8^-THC-COOH, with six samples only positive for Δ^8^-THC-COOH without the presence of Δ^9^-THC-COOH [84]. A summary of the analysis of Δ^8^-THC and its metabolites in different biological matrices is displayed in Table 1.

### 5.5. Stability of Δ^8^-THC and Its Metabolites

A recent study on the stability of Δ^8^-THC and its metabolites; 11-OH-Δ^8^-THC and Δ^8^-THC-COOH in drug-free urine was reported. The study was carried out at different pHs (4.5, 7, and 9) and temperature conditions (4 °C, 20 °C, and 45 °C) for 28 days using an Abbott Architect Plus c4000 Autoanalyzer. The Lin-Zhi enzymatic immunoassay method at the cut-off concentration of 25 ng/mL was used for the analysis. Samples were analyzed daily in the first week and once a week for the remainder of the study or until the target analyte was below the cut-off concentration of 25 ng/mL [85].

The study revealed that the response of Δ^8^-THC varied significantly depending on pH and temperature conditions. At 4 °C, Δ^8^-THC instrument response decreased by about 30% over 7 days, irrespective of pH. Furthermore, no notable declines in response were observed for Δ^8^-THC at 4 °C, regardless of pH, for the remainder of the study. However, at 20 °C, regardless of pH, Δ^8^-THC instrument response decreased by more than 70% over 7 days. By day 14, Δ^8^-THC was undetectable at 20 °C, regardless of pH. Additionally, Δ^8^-THC was not detectable after a single day at 45 °C, regardless of pH [85].

During the period of 7 days, the instrument responses for 11-OH-Δ^8^-THC decreased by 10–15%, 15–20%, and 40–60% at 4 °C, 20 °C, and 45 °C, respectively, regardless of pH. However, at 4 °C, there was no significant decline in the 11-OH-Δ^8^-THC instrument response for the remainder of the study, regardless of pH. By day 14, 11-OH-Δ^8^-THC was not detectable at pH 4.5 and 45 °C. However, at 20 °C and 45 °C, the instrument responses for 11-OH-Δ^8^-THC declined by 10–20% and 40–60%, respectively, for the remainder of the study, regardless of pH. As for Δ^8^-THC-COOH, at pH = 4.5, the instrument response decreased by 40–70% while at pH = 7 and pH = 9, the response decreased by 0–5% regardless of temperature. For the remainder of the study and at pH = 7 and pH = 9, no significant decline in Δ^8^-THC-COOH response was observed [85].

In another stability study, it was suggested that the effectiveness of behavioral modification in rats might depend on the stability of Δ^8^-THC. Specifically, when pure Δ^8^-THC was used, it led to the fastest reduction in avoidance behavior. On the other hand, when partially deteriorated Δ^8^-THC was employed, it did not have a significant impact on avoidance extinction. However, it is important to note that this observation could be influenced by factors like tolerance and the possibility of withdrawal symptoms arising [86]. Inside the human body, Δ^8^-THC was rapidly and extensively metabolized by the liver into the l1-hydroxy metabolite, shedding light on the assumption that the active form of Δ^8^-THC may be the 11-hydroxy metabolites [87].

## 6. Pharmacology of Δ^8^-THC

### 6.1. CB1 and CB2 Activation

The endocannabinoid system comprises the endocannabinoids, cannabinoid receptors, and enzymes responsible for the synthesis and degradation of endocannabinoids [88].

Both cannabinoid binding receptors, CB1 and CB2, are G protein-coupled receptors that are part of the endocannabinoid system interacting and responding to cannabinoids and endocannabinoids [89]. CB1 receptors are highly concentrated in specific brain regions and are less abundant in a more widespread manner, primarily influencing the psychoactive effects of cannabinoids. CB1 receptors affect functions like mood and appetite. Their activation is associated with the psychoactive effects of cannabis. CB2 receptors have a more limited distribution, being located in various immune cells and a small number of neurons. CB2 receptors mainly modulate inflammation and immune cell activity without causing psychoactive effects [88]. Both CB1 and CB2 receptors predominantly couple to inhibitory G proteins, sharing pharmacological influences with other GPCRs. Consequently, the cellular response to specific cannabinoid receptor ligands is intricately shaped by factors such as partial agonism, functional selectivity, and inverse agonism [90]. Both CB receptors play a role in the body’s response to cannabinoids, whether they are produced by the body (endocannabinoids) or those from external sources. The identified eicosanoids such as anandamide (arachidonoylethanolamide), 2-arachidonoylglycerol, and 2-arachidonylglyceryl ether (noladin ether) [91].

Interestingly, Δ^8^-THC exhibits the ability to competitively attach to the orthosteric sites of both CB1 and CB2 receptors, with Ki values falling within the nanomolar (nM) range. Notably, the Δ^8^-THC intriguing effects are not restricted to a specific species, as evidenced by consistent impacts observed across human, rat, and mouse receptors. In detail, Govaerts and colleagues examined binding affinities of Δ^8^-THC in both murine tissues and cultured cells, that were genetically modified to express human cannabinoid receptors. They utilized the GTPγS functional assay to measure the ability of Δ8-THC to activate the G protein-coupled receptors documented binding affinities of 251 nM and 417 nM against CB1 and CB2 receptors, respectively [92]. Husni et al. [32] utilized also the GTPγS functional assay. He and Radwan et al. [27] independently reported affinity values of 78 nM and 12 nM against CB1 and CB2 receptors in human-transfected cells (HEK293), respectively. On the other hand, Nadipuram and colleagues reported close affinity values of 28.5 nM and 25.0 nM against the same receptors in human-transfected cells (HEK293), respectively [93]. Similarly, in their study focused on the synthesis of specific ligands for the CB2 receptor in rats’ brain cortex tissue, Huffman’s research team reported an affinity value for Δ^8^-THC of 44 nM [94]. In contrast, when considering mouse CB1 and CB2 receptors, Δ^8^-THC exhibited binding affinities, as measured by Ki values, within a range of 39.3 nM to 179 nM [92,95,96,97,98,99,100]. The antiemetic effects of 5 mg/kg and 10 mg/kg Δ^8^-THC could be counteracted by a CB1 antagonist, but not by a CB2 antagonist. It is important to highlight that Δ^8^-THC exhibited superior effectiveness in preventing vomiting compared to Δ^9^-THC, despite its lower affinity for CB1 and CB2 receptors [101]. In human-origin stably transfected CHO-K1 cell lines, Δ^8^-THC displayed a dose-dependent inhibition of forskolin-induced cyclic AMP accumulation, demonstrating an EC_50_ of 82 nM [102]. A separate investigation utilizing Rat CHO-K1 (transfected) cell lines observed a similar dose-dependent inhibition of forskolin-induced cyclic AMP accumulation by Δ^8^-THC, revealing an EC_50_ of 27.4 nM [103]. Furthermore, an additional study involving whole brain synaptosomes revealed a distinct U-shaped response in Δ^8^-THC -mediated inhibition of forskolin-induced cyclic AMP accumulation, reaching an Emax of 13% inhibition at 1 μM [104]. These comprehensive findings depict Δ^8^-THC as a versatile modulator of cannabinoid receptors, exerting intricate effects at molecular and cellular levels across diverse biological systems. Its competitive binding to CB1 and CB2 receptors, demonstrated with nanomolar affinity across species, showcases a nuanced molecular engagement.

### 6.2. Neurotransmitters’ Levels and Activities

Several manuscripts have investigated how Δ^8^-THC influences the levels of acetylcholine in order to elucidate its impact on the central nervous system. The effects of Δ^8^-THC on the hippocampus were found to be dose-dependent, and this was linked to alterations in acetylcholine turnover in the striatum through a stereospecificity-affected turnover [105,106]. Conversely, another study did not find convincing proof to suggest any intrinsic pharmacological impact of the Δ^8^-THC on monoamine oxidase (MAO) activity [107]. Noticeable reductions in dopamine and serotonin concentrations were observed in both the hypothalamus and the hippocampus, respectively. Meanwhile, norepinephrine levels tended to rise in both the hypothalamus and hippocampus [108]. Observable reductions in serotonin concentration were also detected in another study upon the uptake of Δ^8^-THC [109]. Δ^8^-THC exposure and stressful environment may lead to increased dopamine reuptake in the striatum, potentially influencing cannabis-induced behaviors under stress [110]. Furthermore, Δ^8^-THC induced a dual-phase impact on the absorption and discharge of dopamine and norepinephrine in both the corpus striatum and hypothalamic areas of the brain [111]. Sagratella and colleagues undertook several investigations on rabbits to examine the potential role of diminished GABA transmission [112]. Other studies have indicated that Δ^8^-THC inhibits adenylate cyclase and subsequently cAMP in a dose-dependent manner, though this mechanism is not thought to be the origin of behavioral alterations [104].

### 6.3. Cognitive Functions

When Δ^8^-THC is administered before a decision-making-based test, it can lead to discriminative behavior, indicating alterations in a state-dependent manner. The study explored the induction and reversal of memory and learning impairment as a means to differentiate between substances, achieved by altering the ‘correct’ choice [113]. Experiments conducted by Jarbe et al. at various doses collectively suggest that Δ^8^-THC enhanced the abilities of rats to pass the T-maze. Individuals subjected to higher doses of Δ^8^-THC required fewer trials to acquire the maze-learned behavior [114,115]. This result was in agreement with another study that identified the discriminative stimulation effects of Δ^8^-THC [116]. However, in contrast, Doty et al. found that rats subjected to phencyclidine (PCP) were responsive to Δ^8^-THC. Although this did not directly assess the potential discriminative effect of Δ^8^-THC itself, these authors did find that Δ^8^-THC altered the dose–response curve of PCP’s discriminative stimulus. The mechanism of this interaction remains unclear and likely involves multiple factors. Interestingly, this is one of the few effects where Δ^8^-THC did not exhibit a clear dose dependency [117].

Evaluating the effects of Δ^8^-THC on pigeons suggested a dose-dependent effect of Δ^8^-THC on pigeons trained with Δ^9^-THC, thus mirroring similar findings reported in rats [118]. Utilizing the fixed behavioral test-resistance to the extinction of avoidance responses in rats, pure Δ^8^-THC led to the quickest cessation of avoidance [86]. Examining different behavioral symptoms, Δ^8^-THC exhibits activity in humans similar to Δ^9^-THC. When assessed through both oral and intravenous administration, it demonstrates a relative potency ratio of 2:3. Doses ranging from 1 to 6 mg of Δ^8^-THC intravenously induce a broad spectrum of effects reminiscent of cannabis [33].

### 6.4. Analgesic and Hypothermic Activities

In a short-term study, dose-dependent pain relief and body temperature decrease were observed in rats after Δ^8^-THC administration. These effects were noticeable within the range of doses of Δ^8^-THC and accompanied by complete tolerance [119]. In the vaporized form, employing the tail withdrawal assay in male rats, a mixture containing three parts CBD to one-part Δ^8^-THC produced an immediate analgesic effect when compared to vapor from the control vehicle [120]. In the same study, immediately after exposure, the mixture induced a reduction in body temperature, leading to a hypothermic effect compared to exposure to the control vehicle [120].

Upon oral ingestion, Δ^8^-THC exhibited an analgesic effect that was dose-dependent when measured by the hot-plate method. Interestingly, this analgesic effect was found to be comparable in potency to Δ^9^-THC, but both were notably less potent than morphine [121]. Conversely, when administered intrathecally rather than orally, Δ^8^-THC was reported to be less potent than Δ^9^-THC employing the tail-flick test [122]. In a combination study, to understand the mechanism of analgesic action, while morphine pretreatment did amplify the analgesic effect, administering naloxone to mice beforehand did not impede it. This suggests that the pain-relieving effects brought about by Δ^8^-THC do not arise from a direct interaction with the opiate receptor [122].

Another study demonstrated the interactions in the use of Δ^8^-THC and opioids within intracellular systems. The potential shared points of interaction include Gi/o protein-coupled receptors. These receptors lead to a decrease in both calcium entry into neurons and the calcium content. This subsequently results in a reduction of cAMP levels and induces hyperpolarization of neurons. These effects are facilitated through the involvement of both ATP- and apamin-sensitive potassium channels [123]. El-Alfy et al. were able to corroborate and establish statistical significance for an existing, though less potent, dose-dependent antinociceptive effect of Δ^8^-THC relative to Δ^9^-THC when assessing for antidepressant-like properties [124]. In a more comprehensive investigation, Welch and his team administered CB1 and CB2 selective antagonist (SR141716A) alongside Δ^8^-THC using the tail-flick test to uncover any potential analgesic role through the cannabinoid receptors. The authors noted varying degrees of antagonism when switching between routes of administration for both SR141716A and Δ^8^-THC. This implies a potentially intricate role of cannabinoid receptors involving various subtypes and distinct localization throughout the central nervous system [125].

### 6.5. Antiepileptic Activities 

A statistically significant dose-dependent reduction in audiogenic response score, indicating diminished seizure activity, was observed in female rats at all three doses of Δ^8^-THC administered with an effective dose for complete seizure inhibition of 6.5 mg/kg, exceeding the upper limit of the trial protocol [126]. Another study explored diverse methods, including kindling to provoke seizures, a higher intraperitoneal dose of 10 mg/kg of Δ^8^-THC was required to significantly reduce seizure manifestations, with significant outcomes defined as >25% protection of the population [127]. In a different experimental animal model, seizures were induced in Senegalese baboons through phototherapy and kindling. Observations revealed dose-dependent prevention and reduction of kindled seizures from electrical stimulation, but no such effect was observed with light-induced seizures after intraperitoneal injections of Δ^8^-THC [128].

The same research group once again employed kindling and a similar seizure staging model, this time assessing another animal model: cats. In contrast to their previous findings, Δ^8^-THC demonstrated no significant effect on suppressing or reducing seizure behavior during any stage of kindling in cats at doses ranging from 0.5 to 4 mg/kg [129]. An alternative study utilized cobalt-epileptic rats observing the activity at the epileptic focus, suggesting that the potential anticonvulsant activity of Δ^8^-THC may not be consistent across different types of seizures and could potentially exacerbate convulsant activity in seizures arising from other causes [130]. Interestingly, both Δ^8^-THC and Δ^9^-THC were determined to have similar relative potency in anticonvulsant activity for PTZ-induced seizures in mice, contrary to previous studies in rats. Additionally, the same study showed that Δ^8^-THC exhibited an additional protective effect against seizures when combined with phenobarbital, but not with diphenylhydantoin [131].

### 6.6. Cardiovascular Activities

In a short-term study, dose-dependent effects on the reduction of heart rate were observed in rats while complete tolerance to the decrease in heart rate was established within only 13 days [119]. In a conflicting study, Gong and colleagues conducted a three-part experiment examining bronchodilation and tolerance to Δ^8^-THC. They observed dose-dependent responses. These responses included statistically significant inducible bronchodilation and tachycardia [132]. Smiley KA observed arrhythmia and reduced contractility on perfused rat hearts [133]. A subsequent in vitro study in rats investigated changes in contractility by measuring nitric oxide production recording a significant reduction [134]. Another study suggested alterations in the activity of monoamine oxidase enzymes in the hearts of live rats as a potential mechanism for bronchodilation and tachycardia. An early study that compared intravenous and intra-arterial Δ^8^-THC in rats mentioned a potential role as a peripheral vasoconstrictor with intra-arterial administration, while the intravenous administration led to immediate increases in blood pressure, followed by subsequent drop and a decrease in heart rate [135].

### 6.7. Gastro-Intestinal Tract Activities

Δ^8^-THC, when administered orally, led to dose-dependent reductions in intestinal motility in mice. This effect was assessed by monitoring the transit of a charcoal meal through the digestive tract. Specifically, as the dosage of Δ^8^-THC increased, there was a corresponding decrease in the pace at which the charcoal meal moved through the intestines [121]. In a study targeted the food intake, a very low dose of Δ^8^-THC (0.001 mg/kg) caused increased food consumption and a tendency to improve cognitive function, without cannabimimetic side effects suggesting that the low dose might serve as a potential therapeutic agent in the treatment of weight disorders [107]. Inversely, rats experienced weight loss, decreased intake, bradycardia, and decreased body temperature after administration of Δ^8^-THC in another study; however, partial or complete tolerance developed to many of these physiologic changes [132]. Emesis prevention in radiation-treated shrews showed a statistically significant dose-dependent response at single doses of 5 mg/kg and 10 mg/kg of Δ^8^-THC. The prevention of emesis could be reversed with a CB1, but not with a CB2 antagonist. It is noteworthy that Δ^8^-THC demonstrated greater efficacy in preventing emesis compared to Δ^9^-THC, despite having a lower binding affinity for CB1 and CB2 receptors presenting a unique scenario [101]. In another study on emesis prevention using oral Δ^8^-THC administration every 6 h for 24 h in eight children undergoing multiple cycles of antineoplastic regimens for hemolytic malignancies over 8 months, the study found that administering Δ^8^-THC over 24 h effectively prevented emesis on the day of treatment and for the subsequent two days [136]. On the other hand, a recent retrospective case report suggests that frequent oral administration of Δ^8^-THC may be associated with cannabinoid hyperemesis syndrome indicating a paradoxical, dose-dependent, and time-dependent effect. The factors that could be involved in such results include a potential biphasic or antagonist effect at higher doses, in addition to the receptor downregulation and subsequent alterations in the endocannabinoid system [73].

### 6.8. Anticancer Activities 

Several scientific in vitro and in vivo studies have explored the potential anti-cancer effects of Δ^8^-THC. These investigations aim to understand how this compound may impact different kinds of cancer cells and their proliferation. To provide a clear overview of these findings, a selection of studies has been compiled and organized in the following Table 2 for easy reference and analysis.

The diverse array of studies outlined in Table 2 collectively underscores the multifaceted antitumor properties of Δ^8^-THC. The findings consistently reveal its ability to impede critical processes such as DNA synthesis, cancer cell growth, and cellular respiration across a spectrum of cell types, including murine leukemia cells, human cancer cells, and lymphoma cells. Notably, Δ^8^-THC demonstrates promise in inhibiting multidrug resistance in certain cancer cells and induces a range of cellular responses, from apoptosis and autophagy to modulation of molecular markers associated with cell cycle progression. Furthermore, in vivo experiments on mice carrying lung adenocarcinoma suggest that Δ^8^-THC may not only retard tumor growth but also enhance the life span of treated animals. These cumulative results support the potential therapeutic relevance of Δ^8^-THC in combating cancer through diverse mechanisms, warranting further exploration and clinical investigation.

### 6.9. Immunomodulatory Activities

Numerous scientific investigations have focused on understanding the immunomodulatory properties of Δ^8^-THC. These studies aim to unravel how this compound can influence the immune system. To facilitate a comprehensive grasp of these findings, a subset of these studies has been thoughtfully consolidated and presented in Table 3 for easy reference and analysis.

In summary, the diverse array of studies examining the effects of Δ^8^-THC across different cells, organs, and systems underscores its multifaceted immunomodulatory impact. Δ^8^-THC demonstrates a range of outcomes, from inhibiting lymphocyte proliferation and inducing cell death in macrophages to influencing immune responses in vivo. Notably, it exhibits both immunosuppressive effects, as observed in the reduction of hemolytic plaque-forming cells and the suppression of autoimmune encephalomyelitis, and modulatory effects on behavior, evidenced by the regulation of lever-pressing behavior through intricate signaling pathways in the brain. Additionally, Δ^8^-THC appears to antagonize oxidative stress, suggesting potential protective effects on B lymphocytes and fibroblasts. The presented findings collectively highlight the complex and context-dependent nature of Δ^8^-THC’s interactions within the biological systems studied, emphasizing the need for further research to elucidate its therapeutic immunomodulatory potential and potential adverse effects.

### 6.10. Ocular Activities

The mechanism of Δ^8^-THC in corneal injury treatment was found to be mediated by cannabinoid receptors, as a study observed reduced pain and inflammation after corneal injury in mice with topical administration of Δ^8^-THC [156]. Interestingly, additional lipid-based vehicles for topical preparation of different Δ^8^-THC-containing topical formulations enhanced corneal penetration and minimized systemic absorption [157]. Similar results were obtained after the application of a submicron emulsion as an ocular vehicle for Δ^8^-THC as a treatment for intraocular pressure in rabbits [158]. An early study involving eight rabbits observed a decrease in the intraocular pressure following intravenous administration of Δ^8^-THC, but this effect was not dose-dependent [159]. In another study, topical Δ^8^-THC did eventually lead to a reduction in intraocular pressure in rabbits [160]. Further studies by the same team indicated that the effect was likely not centrally mediated or a result of the drug crossing the blood–brain barrier [161].

### 6.11. Locomotor Activities

In the vaporized form, a concentration of 10 mg of Δ^8^-THC initially induced hyperlocomotion in male rats within the first 30 min of the session, followed by a hypo locomotor effect later in the session [120]. In mice, both Δ^8^-THC and Δ^9^-THC demonstrated a notable decrease in spontaneous activity, leading to reduced awareness. They also caused a loss of muscle coordination, diminished sensorimotor responses, and elicited a hunched posture and gait reminiscent of the effects induced by narcotic analgesics. Additionally, when administered intravenously, both Δ^8^-THC and Δ^9^-THC exhibited a quicker onset of activity compared to intracerebral administration [87].

### 6.12. Fertility Affecting Activities

In an investigation into its anti-androgenic properties, Δ^8^-THC demonstrated remarkable effectiveness as an inhibitor of mitochondrial O_2_ consumption within human sperm cells. This finding underscores the significant impact that Δ^8^-THC can exert on the metabolic processes crucial for sperm function and reproductive physiology [162]. In another study, rats that were exposed chronically to Δ^8^-THC displayed significantly reduced body weight and lower levels of follicle-stimulating hormone and luteinizing hormone in comparison to the control group. However, it is worth noting that these hormone levels did recover once the drug was discontinued. The precise clinical implications of these effects on fertility remain uncertain and warrant further investigation [163].

### 6.13. Antidepressant Activities 

The antidepressant effects of Δ^8^-THC were evaluated, using the automated mouse-forced swim and tail suspension tests. Interestingly, doses of 1.25, 2.5, and 5 mg/kg administered intraperitoneally demonstrated a U-shaped dose-response in terms of their antidepressant action [124].

### 6.14. Toxicity of Δ^8^-THC

Following the receipt of 104 reports detailing adverse events in individuals who had consumed Δ^8^-THC products between December 1, 2020, and February 28, 2022, the Food and Drug Administration (FDA) released a report underscoring the potential hazards linked with Δ^8^-THC products and emphasized the necessity for regulatory oversight and public awareness. The FDA report highlights five major significant concerns regarding Δ^8^-THC products [38]. Firstly, these products have not undergone FDA assessment or approval for safe usage, potentially posing risks to public health. There are worries about inconsistent formulations, incorrect labeling, and varying Δ^8^-THC levels. Some products may be misleadingly labeled as “hemp products”, potentially leading consumers to underestimate their psychoactive effects. Secondly, the report expresses apprehension about products claiming therapeutic benefits without FDA approval. This could endanger consumers, as the safety and efficacy of such products have not been confirmed. The data presented indicates possible adverse events, particularly among pediatric patients, emphasizing the need for caution. Thirdly, Δ^8^-THC shares similar psychoactive effects with Δ^9^-THC, indicating a comparable level of impairment. Fourthly, concerns are raised about the manufacturing process, which may involve potentially harmful chemicals and lead to contaminants in the final product. Finally, manufacturers’ packaging Δ^8^-THC products in ways appealing to children is a notable concern, as this could lead to unintentional exposure [38].

On 14 September 2020, the Centers for Disease Control and Prevention (CDC) announced an alert, warning healthcare workers and the public about the recent rise in adverse health events related to Δ^8^-THC consumption. It highlights that emergency room visits mentioning Δ^8^-THC as the primary complaint were mostly concentrated in southern states without legal adult-use marijuana access. In these cases, Δ^8^-THC products are often marketed as hemp products, potentially leading to accidental intoxication due to consumer confusion. The CDC advises caution, noting that Δ^8^-THC products can be intoxicating, may have misleading labels, and should be kept away from children [164].

In the same context, chemists expressed serious concerns over impurities in Δ^8^-THC, as many Δ^8^-THC products available may not be entirely pure, containing other cannabinoids in addition to uncertain by-products with unclear health effects. The lack of oversight and testing raises safety worries. Regulators are struggling to manage Δ^8^-THC’s sales and chemists stress the importance of enhanced regulation and transparency in production. They also caution against potential risks stemming from unregulated synthesis methods [165]. In light of this, the findings of a recent study combining techniques such as NMR, HPLC, and MS revealed that the tested products contain not only 10 different Δ^8^-THC related structures but also several impurities, present in concentrations significantly higher than what is stated on the certificates of analysis for these products [66].

A recent investigation into self-reported effects linked to Δ^8^-THC revealed similarities to those reported during acute cannabis intoxication. The most commonly reported adverse events were related to psychiatric disorders, followed by respiratory, thoracic, and mediastinal issues, as well as nervous system problems. The most frequently cited symptoms in these reports included feelings of anxiety, coughing, and paranoia [166]. This study supported a prior investigation that detailed cases of two individuals, aged 19 and 20, experiencing acute psychiatric distress after consuming a Δ^8^-THC product. These cases demonstrate a conceivable link in time between the consumption of products containing Δ^8^-THC and the emergence of manic or psychotic symptoms, implying a potential relationship between dosage and response [167]. In pediatric cases, the clinical manifestations resulting from exposure to Δ^8^-THC encompass a spectrum of effects, which extend beyond but are not confined to alterations in mental status. These include presentations such as episodes reminiscent of seizures and irregularities in vital signs [168].

In an early assessment of tolerance to Δ^8^-THC, after chronic administration and mice sacrifice, the livers demonstrated significant increases in tyrosine aminotransferase activity with acute administration; however, little-to-no increased activity was noted after 8 and 12 weeks of chronic administration of Δ^8^-THC [169]. Additionally, the Birmingham School of Medicine has reported recently a notable surge in the frequency of emergency department admissions, both in their own hospital and in several others, attributed to the utilization of Δ^8^-THC [170]. In a study on the antiemetic effects of oral Δ^8^-THC in eight children undergoing hemolytic malignancies over an 8-month period, the authors observed minimal incidence of anxiolytic or psychotropic effects, taking into consideration the challenges in both identifying and self-reporting these symptoms in young children [136].

### 6.15. Tolerance

It is worth noting that tolerance to the biological effects of Δ^8^-THC has been a recurring observation in several scientific publications. These studies have consistently pointed out that over time, the response to Δ^8^-THC can undergo significant alterations. This phenomenon has been extensively documented, shedding light on the complex nature of how the body interacts with this compound. Mice became completely tolerant to the hypothermic effects and partially tolerant to extended phenobarbital-induced sleeping times and catalepsy within 38 days of daily intravenous Δ^8^-THC administration [171,172]. Moreover, Δ^8^-THC -tolerant mice were cross-tolerant to the body temperature-lowering effects of chlorpromazine, but not to morphine or pentobarbital [173,174]. When considering morphine interactions, a study did find that morphine-tolerant mice treated with Δ^8^-THC induced heightened catalepsy rather than cross-tolerance [175]. Furthermore, the development of cross-tolerance to the prolongation of pentobarbital-induced sleep by Δ^8^-tetrahydrocannabinol and 11-hydroxy-Δ^8^-tetrahydrocannabinol was reported in mice [172]. In a study exploring the reducing power of Δ^8^-THC against seizure manifestations, chronic administration of intraperitoneal 15 mg/kg Δ^8^-THC rapidly led to tolerance in preventing seizure activity. Only two intraperitoneal injections were required to induce tolerance [127]. At elevated levels of Δ^8^-THC, the ability to counteract epileptic activity seems to diminish in Senegalese baboons following neural discharge at the targeted stimulation site [128]. In a short-term study regarding the cardiovascular system, a relationship between the administered dose (2 mg/kg) of Δ^8^-THC and a decrease in heart rate was evident. Notably, a complete tolerance to the heart rate reduction occurred in just 13 days [118]. The immunomodulatory effects of 60 mg/kg of Δ^8^-THC against direct hemolytic plaque-forming cells in the mice spleen showed a state of reduced responsiveness (hyporesponsiveness) developed when a pretreatment regimen was employed [150]. 

### 6.16. Withdrawal Activities

Some research articles have focused on the potential of Δ^8^-THC withdrawal. In general, the decrease in exploratory behavior with short-term and long-term administration of Δ^8^-THC supports the CNS depressant effects. Significant changes in rat’s defecation (potentially indicative of mood changes) and grooming behaviors (potentially indicative of withdrawal symptoms) were noted [176]. In morphine-dependent mice, increased jumping and forepaw tremors upon administration of naloxone were considered consistent with withdrawal. The administration of Δ^8^-THC was able to inhibit these behavioral modifications. Also, high doses of Δ^8^-THC induced hyperactivity in morphine-dependent mice rather than ataxia/catalepsy [175].

The upregulation of the CB1 receptors coupled with the mitigation of withdrawal symptoms with CB1 stimulation indicates that dysregulation of the endocannabinoid system may have a role in opiate-induced withdrawal [175]. A contradictory study showed that a single administration of 10 mg/kg of Δ^8^-THC was insufficient to suppress naloxone-induced morphine withdrawal [177]. Abrupt cessation in rats after chronic administration of Δ^8^-THC for 30 days increased spontaneous yawning in a similar way to that of opioid withdrawal [178]. On the other hand, the acute administration portion of the study identified that Δ^8^-THC reduced yawning induced by dopaminergic and cholinergic agonists [178]. Further investigations into the potential dopaminergic modulation of Δ^8^-THC may clarify any potential for inducing overuse-associated behaviors.

Modulation of the arachidonic acid pathway and the stimulation of the prostaglandin EP3 receptors may also be involved in dependence and withdrawal [178]. Furthermore, with this induced prostaglandin synthesis, the rats’ brain reduces natural production and becomes dependent on THC for prostaglandin maintenance [178]. The injection of 10 mg/kg diclofenac, an inhibitor of prostaglandin synthesis, intraperitoneally 30 min before the synthetic CB1 antagonist, SR 141716A, induced withdrawal symptoms in mice that were naïve to cannabinoids. These signs closely resembled those of Δ^8^-THC withdrawal symptoms. Conversely, the administration of 10 mg/kg Δ^8^-THC i.p., 15 min before SR 141716A, prevented the manifestation of these withdrawal signs. These findings imply that abrupt cessation of Δ^8^-THC may cause a decrease in prostaglandin levels that contributes to the withdrawal syndrome [179]. Anggadiredj and colleagues’ study suggested that the potential for Δ^8^-THC to provoke withdrawal symptoms is not yet well understood and warrants further research [179].

As of now, there is no approved medication widely and consistently proven effective for cannabinoids dependence and withdrawal [180,181]. Explored medications include those alleviating symptoms of cannabinoids withdrawal (such as dysphoric mood and irritability), those influencing endogenous cannabinoid receptor function, and those successful in treating other substance abuse or psychiatric conditions [182]. Among these, some drugs such as buspirone demonstrated efficacy against cannabinoid dependence in controlled clinical trials [183]. Also, nabiximols, in a randomized clinical trial, were utilized as a successful agonist replacement therapy against cannabinoid withdrawal [184]. In two case reports, it was demonstrated that dronabinol effectively diminished withdrawal symptoms and alleviated the subjective effects associated with cannabinoids [185]. Recent pre-clinical studies also highlight the potential of fatty acid amide hydrolase inhibitors like URB597, endocannabinoid-metabolizing enzymes, and nicotinic alpha7 receptor antagonists such as methyllycaconitine [186].

### 6.17. Clinical Trials on Δ^8^-THC

Resources discussing the clinical trials on Δ^8^-THC are very limited; however, most of its potential benefits are dependent on marketing claims. The first clinical study was conducted in 1973 on six participants for the comparison between Δ^8^-THC and Δ^9^-THC following orally and intravenously administered doses [33]. The study participants were orally administered drug-containing chocolate cookies containing 20 mg and 40 mg of Δ^8^-THC and 20 mg Δ^9^-THC. The trials were conducted over three weeks intervals. Both Δ^8^-THC and Δ^9^-THC clinical effects were evaluated by a narrative log covering every 30 min for up to 5 h with the records of pulse rate, blood pressure, and conjunctival color. The study results showed that all three treatments produced similar somatic, psychic, and perceptual effects. The lower Δ^8^-THC dose (20mg) produced the least clinical effects with slower onset and shorter duration of action [33].

For intravenous administration, both Δ^8^-THC and Δ^9^-THC solutions in 95% Ethanol (1 mg per 0.20 mL) were injected into a normal saline solution after the start of dripping. In a single experiment, three volunteers received Δ^8^-THC, while four subjects received Δ^9^-THC. Δ^8^-THC was administered to the three subjects in a total of six separate doses. The participants experienced the same qualitative symptoms of Δ^9^-THC. Moreover, the intensity and duration of clinical effects produced by the treatment with Δ^8^-THC were dose-dependent. Generally, the study showed that Δ^8^-THC produced slightly weaker effects than Δ^9^-THC. The authors concluded that both Δ^8^-THC and Δ^9^-THC produce the same clinical effects when administered intravenously, however, Δ^8^-THC exhibited approximately 2/3 the potency of Δ^9^-THC, when administered orally [33].

In 1995, Mechoulam’s group conducted a small clinical trial on Δ^8^-THC in children, aged 3–13 years with different hematologic cancers. Eight pediatric cancer patients who were expected to vomit upon receiving an antineoplastic treatment received an oral dose of Δ^8^-THC oil solution at a dose of 18 mg/m^2^ two hours before each chemotherapy session. The selected subjects were monitored for up to 8 months with a total number of 480 treatments. Complete prevention of vomiting was achieved with almost no side effects upon the use of Δ^8^-THC treatments, regardless of the antineoplastic protocol followed [136].

In conclusion, more clinical research is required to determine the potential benefits of Δ^8^-THC; however, legal concerns make these types of studies challenging.

## 7. Limitations

The available data on Δ^8^-THC lack substantial clinical information or comprehensive outcomes regarding its medical applications. Additionally, the analysis of by-products resulting from the synthesis of Δ^8^-THC is still limited, necessitating further investigation into the potential implications of these derivatives. The studies also fall short in providing an in-depth exploration of potential adverse effects and safety concerns associated with Δ^8^-THC and its by-products. Furthermore, the absence of information on long-term studies and chronic effects of Δ^8^-THC use is a notable limitation, impeding a comprehensive understanding of the compound’s sustained impact. Moreover, regulatory and legal considerations related to the production, distribution, and use of Δ^8^-THC are significant obstacles, creating a gap in the overall assessment of this compound.

## 8. Conclusions

This review has provided a comprehensive exploration of the chemical, analytical, and pharmacological aspects of Δ^8^-THC. The review shed light on its emergence as a psychoactive cannabinoid with increased popularity, driven by its superior stability and the easy synthetic procedure compared to the more well-known Δ^9^-THC. The paper delved into the methods of extraction, purification, and structure elucidation of Δ^8^-THC, addressing the crucial question of whether it is a natural compound or an artifact. Additionally, strategies for the chemical synthesis and analyses of Δ^8^-THC were presented, along with considerations of potential impurities associated with synthetic manufacturing. The discussion on methods of analysis and detection of impurities in marketed products serves as a valuable guide for researchers, industry professionals, and regulatory bodies. The pharmacological effects of Δ^8^-THC were thoroughly discussed, focusing on its interaction with CB1 and CB2 cannabinoid receptors, as well as other physiological targets. This in-depth analysis offers a comprehensive understanding of the specific effects elicited by Δ^8^-THC. As the interest in Δ^8^-THC continues to grow, this review may serve as a valuable resource for researchers, practitioners, and policymakers in cannabis-related fields. Further research and ongoing scrutiny are warranted to better understand the compound’s properties, implications, and potential applications.

## Figures and Tables

**Figure 1 molecules-29-01249-f001:**
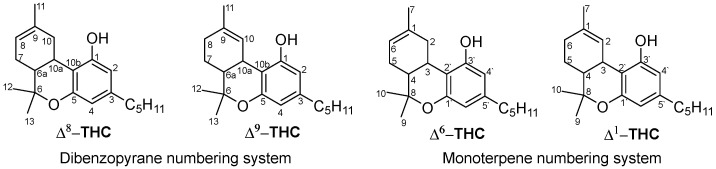
Numbering systems of tetrahydrocannabinols according to the monoterpene (past) and dibenzopyrane (current) numbering systems.

**Figure 2 molecules-29-01249-f002:**
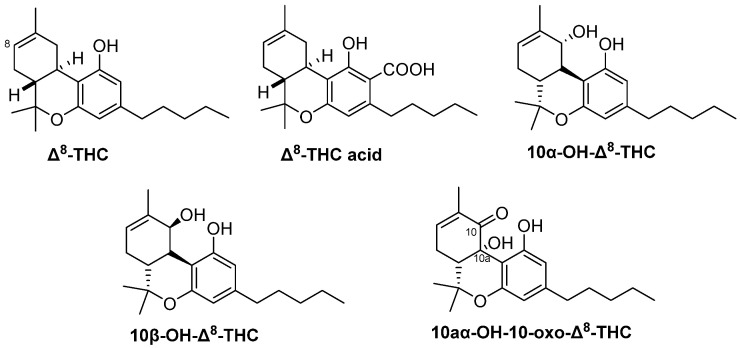
Chemical structure Δ^8^-THC and its derivatives isolated from cannabis.

**Figure 3 molecules-29-01249-f003:**
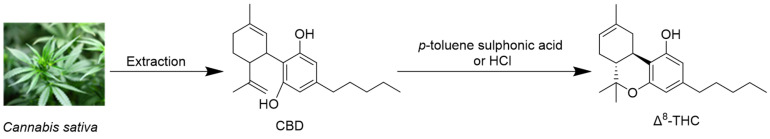
The synthetic pathway of Δ^8^-THC.

**Figure 4 molecules-29-01249-f004:**
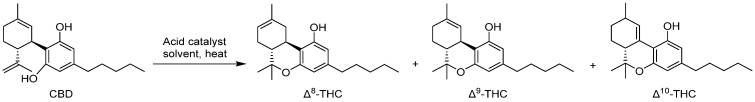
Synthesis of Δ^8^-THC from CBD (yielding small amounts of Δ^9^-THC and Δ^10^-THC).

**Figure 5 molecules-29-01249-f005:**
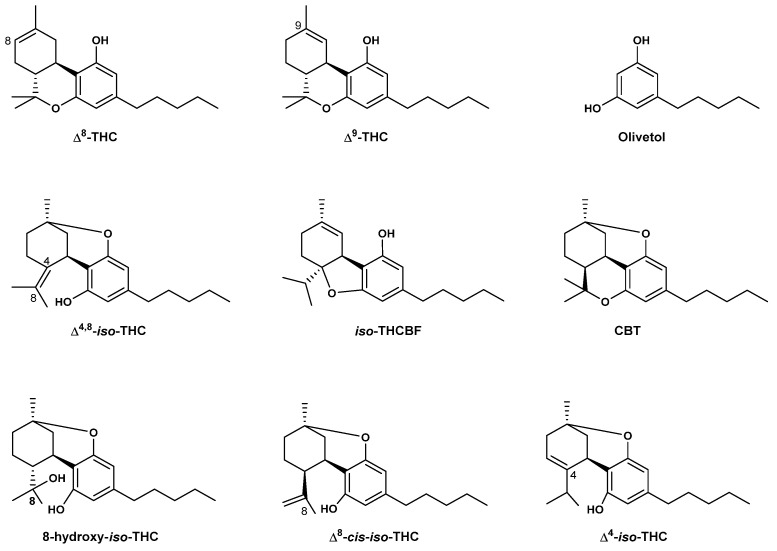
Chemical structures of Δ^8^-THC and its related impurities found in Δ^8^-THC commercial products.

**Figure 6 molecules-29-01249-f006:**
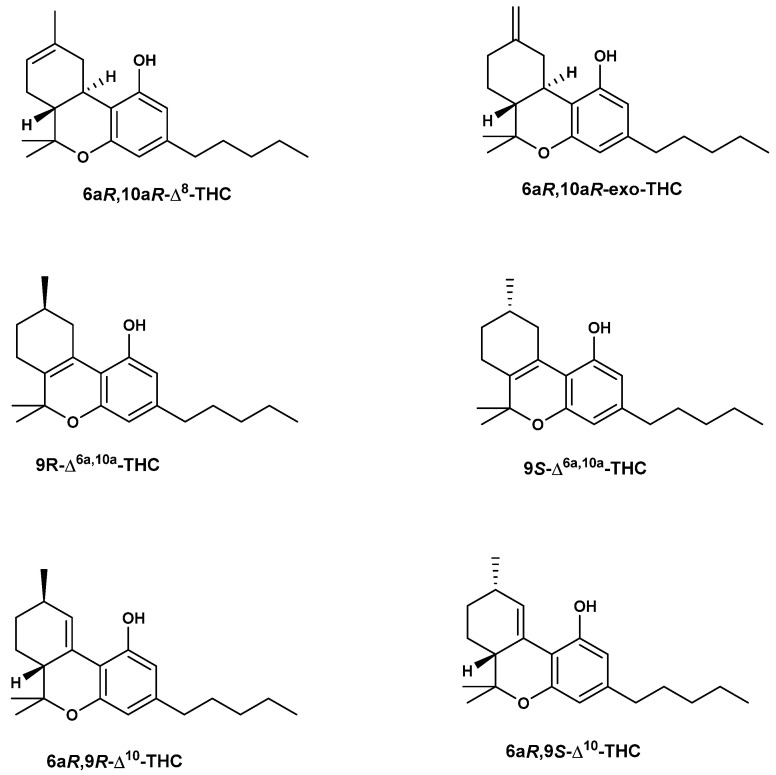
The chemical structures of tetrahydrocannabinol (THC) isomers [67].

**Figure 7 molecules-29-01249-f007:**
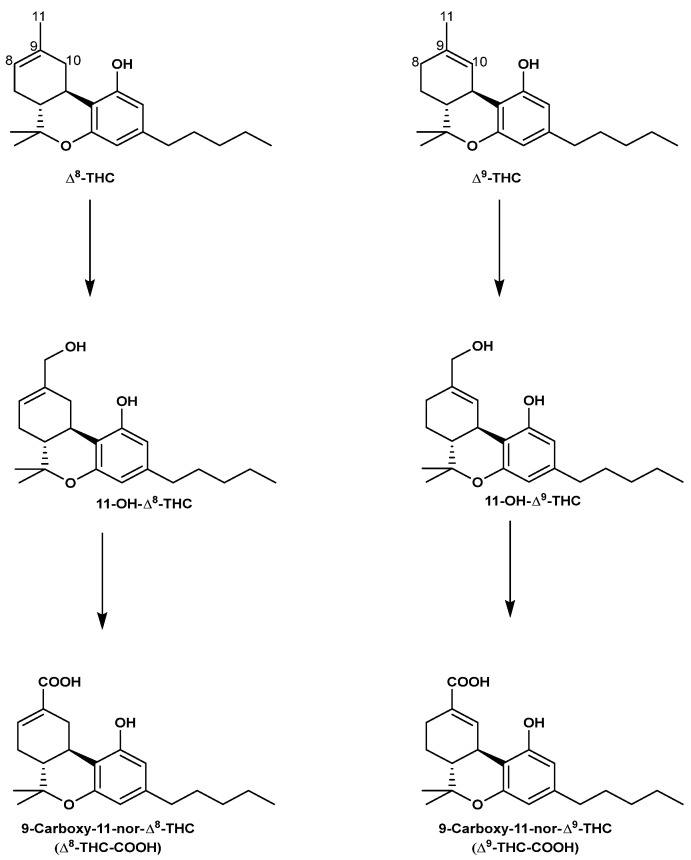
The metabolic pathway of Δ^8^-THC and Δ^9^-THC.

**Table 1 molecules-29-01249-t001:** Analysis of Δ^8^-THC and its metabolites in different biological matrices.

Matrix	Analytical Technique (Analyte)	Sample Preparation	MRMs	Internal Standard (IS)	Reference
Oral fluid	HPLC-MS/MS (Δ^8^-THC)	SPE: I 0 µL of IS + 400 µL of 2% phosphoric acid + 400 µL of oral fluid samples, washed with 400 µL of water/methanol (95:5, *v*/*v*) and eluted by 400 µL of acetonitrile/methanol (90: 10, *v*/*v*)	Quantifier 315.1 → 123.0 (*m*/*z*) a qualifier 315.1 → 135.1 (*m*/*z*)	d_9_-Δ^8^-THC, d_3_-Δ^9^-THC, d_3_-CBD and d_3_-CBN. d_9_-Δ^8^-THC: 324.21 → 123.1 (*m*/*z*)	[79]
Whole blood and serum	LC-ESI-MS/MS (Δ^8^-THC-COOH)	200 µL blood samples + 20 µL of IS + 600 µL of ACN, mixing, and centrifugation, evaporated to dryness+ reconstituted in 200 µL of ACN/H_2_O/Formic acid, 60/40/0.1: *v*/*v*/*v*	(*m*/*z*), 345 → 327 (quantifier), 345 → 299 (qualifier)	THC-d_3_ and 11-OH-THC-d_3_, 10ng, THC-COOH-d_3_	[82]
Human urine	GC/MS (Δ^8^-THC-COOH)	after derivatization and Cannabinoid immunoassay	*m*/*z* 488 → 473, 371 + ve (SIM).	Δ^9^-THC-COOH	[83]
Postmortem urine	GC/MS (Δ^8^-THC-COOH)	SPE after alkaline hydrolysis and derivatization using BSTFA with I% trimethylchlorosilane	(Δ^9^-THC-COOH: *m*/*z*, 371,473 and 488; Δ^8^-THC-COOH: *m*/*z* 488, 473, and 432)	d_9_-Δ^9^-THC-COOH: (*m*/*z* 374, 476, and 491)	[84]

**Table 2 molecules-29-01249-t002:** Anticancer activities of Δ^8^-THC.

Examined Cells, Organs, or System	Type of Study	Results	Reference
Mice, Lewis lung cells, L1210 leukemia cells, and bone marrow cells	In vitro and In vivo	Δ^8^-THC showed a dosage-dependent reduction in DNA synthesis.	[137]
L1210 murine leukemia.	In vivo	Among the compounds examined, Δ^8^-THC exhibited the highest potency, with a remarkable 99% inhibition of DNA synthesis.	[138]
Mice and L1210 murine leukemia	In vivo and in vitro	Δ^8^-THC didn’t Inhibit cancer cells’ respiration.	[139]
Human cells which metabolize polycyclic hydrocarbon carcinogens	In vitro	Δ^8^-THC caused a dose-dependent inhibition of cancer cell growth in addition to a dose-dependent inhibition of [3H]thymidine, [3H]uridine and [3H]leucine incorporation.	[140]
Neuroblastoma cell membranes	In vitro	Δ^8^-THC inhibited adenylate cyclase in plasma membranes	[141]
Human mdr1-gene transfected mouse lymphoma cells	In vitro	Δ^8^-THC exhibited membrane-associated antitumor effects and reversal of multidrug resistance.	[142]
Human oral Tu183 cancer cells	In vitro	Δ^8^-THC exhibited dose-dependent potent inhibition against cancer cellular respiration	[143]
Human oral cancer cell	In vitro	Δ^8^-THC promoted apoptosis and autophagy. Furthermore, it hindered cell migration and invasion. It decreased the production of reactive oxygen and increased levels of glutathione and its expression. it downregulated the expressions of cyclin D1, p53, NOXA, PUMAα, and DRAM, but upregulated the expressions of p21 and H2AX.	[144]
Mice carrying Lewis lung carcinoma	In vivo	Δ^8^-THC led to a dose-dependent retardation of tumor growth. Δ^8^-THC increased the life span of the treated mice and decreased primary tumor size.	[145,146]

**Table 3 molecules-29-01249-t003:** Immunomodulatory activities of Δ^8^-THC.

Examined Cells, Organs, or System	Study Type	Results	Reference
Lymphocytes	In vitro	Δ^8^-THC led to a dose-dependent inability of lymphocytes to integrate the [3H] thymidine.	[147]
T and B lymphocyte	In vitro	Δ^8^-THC inhibited the mitogen-induced T and B lymphocyte proliferation.	[148]
Mouse macrophage J774-1 cells	In vitro	Δ^8^-THC induced cell death of J774-1 cells in a concentration- and/or exposure time-dependent manner. Associated with vacuole formation, chromatin condensation, cell swelling, and nuclear fragmentation.	[149]
BALB/c mice, Plasma and Spleen	In vivo	Δ^8^-THC demonstrated a notable inhibition of direct hemolytic plaque-forming cells in the spleen on day 4.	[150]
Guinea pig, Skin	In vivo	moderate (Grade III) sensitizers, causing allergic contact dermatitis.	[151]
Rats, Brain	In vivo	Suppression of autoimmune encephalomyelitis by Δ^8^-THC was noticed and attributed to its influence on the secretion of corticosterone.	[152]
Rats, Brain	In vivo	Δ^8^-THC reduced lever-pressing behaviour by activating the arachidonic acid cascade, leading to an elevated production of prostaglandin E2 in the brain through the CB1 receptor.	[153]
Rats, Brain	In vivo	Δ^8^-THC suppressed lever-pressing behaviour through the activation of the prostanoid EP3 receptor by elevation of prostaglandin E2.	[154]
Human B-lymphoblastoid and mouse fibroblast cell	In vitro	Δ^8^-THC showed antioxidative effect by preventing serum-deprived cell death that induced by anhydroretinol.	[155]

## Data Availability

Not applicable.

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
