# Peer review of "Chemistry and Pharmacology of Delta-8-Tetrahydrocannabinol"

_molecules, 2024, doi:10.3390/molecules29061249_

Round 1

Reviewer 1 Report

Comments and Suggestions for Authors

The manuscript is a comprehensive review of the various psychotropic elements within cannabis. Specifically, the authors focused on the Δ8-THC derivative. This derivative of THC is very popular due to its synthetic simplicity and lack of regulation. This manuscript provides an excellent analytical approach to exploring potential species of synthetic THC derivatives. The authors accurately describe each isoform and describe various analytical techniques.

Furthermore, the authors also raise an interesting point on the chemical and biological conversion of cannabinoids via regular routes of administration. Furthermore, the authors cover the various effects on a user and potential toxicity due to heavy usage. There were no significant issues in the writing or flow of the manuscript.

I agree that this review could serve as a potentially good outline of the Δ8-THC derivative and its effect on the population and should be published with little changes.

Comments on the Quality of English Language

Overall, I did not see any glaring issues with the English language.

Author Response

Reviewer 1

Comments and Suggestions for Authors

The manuscript is a comprehensive review of the various psychotropic elements within cannabis. Specifically, the authors focused on the Δ8-THC derivative. This derivative of THC is very popular due to its synthetic simplicity and lack of regulation. This manuscript provides an excellent analytical approach to exploring potential species of synthetic THC derivatives. The authors accurately describe each isoform and describe various analytical techniques.

Furthermore, the authors also raise an interesting point on the chemical and biological conversion of cannabinoids via regular routes of administration. Furthermore, the authors cover the various effects on a user and potential toxicity due to heavy usage. There were no significant issues in the writing or flow of the manuscript.

I agree that this review could serve as a potentially good outline of the Δ8-THC derivative and its effect on the population and should be published with little changes.

We would like to thank the reviewer for the nice comments.

Reviewer 2 Report

Comments and Suggestions for Authors

Interesting paper. A smart analysis on the characteristics of Delta-8-tetrahydrocannabinol was made in this review, starting from its origins, chemical routes, pharmacological effects and a hard to answer question related to its nature. Ways to analyze impurities related to the compound were described and a full discussion of the techniques used in literature is present. The manuscript is well organised and understandable. I'm not an English native reader so I can not assess the quality of the English in the paper, but I've not met evident mistakes. The number and the kind of references are adequate to support all the sentences made in the text. For these reasons, from my perspective, the manuscript can be accepted on Molecus in the present form

Author Response

Comments and Suggestions for Authors

Interesting paper. A smart analysis on the characteristics of Delta-8-tetrahydrocannabinol was made in this review, starting from its origins, chemical routes, pharmacological effects and a hard to answer question related to its nature. Ways to analyze impurities related to the compound were described and a full discussion of the techniques used in literature is present. The manuscript is well organised and understandable. I'm not an English native reader so I can not assess the quality of the English in the paper, but I've not met evident mistakes. The number and the kind of references are adequate to support all the sentences made in the text. For these reasons, from my perspective, the manuscript can be accepted on Molecus in the present form

We would like to thank the reviewer for the nice comments.

Reviewer 3 Report

Comments and Suggestions for Authors

The manuscript on the chemistry and pharmacology of delta-8-tetrahydrocannabinol is well written and gives a comprehensive overview of the topic.  It is interesting and provides excellent information on the chemistry and analysis of delta-8-tetrahydrocannabinol.  It was observed that the pharmacological activities were mainly described by referring to pre-clinical studies in animals.  It would be more useful if proof of activity was established in human clinical trials.  The only criticism is that very limited information on clinical trials are provided.  This may be due to the limited availability of clinical data.

Author Response

The manuscript on the chemistry and pharmacology of delta-8-tetrahydrocannabinol is well written and gives a comprehensive overview of the topic.  It is interesting and provides excellent information on the chemistry and analysis of delta-8-tetrahydrocannabinol.  It was observed that the pharmacological activities were mainly described by referring to pre-clinical studies in animals.  It would be more useful if proof of activity was established in human clinical trials.  The only criticism is that very limited information on clinical trials are provided.  This may be due to the limited availability of clinical data

We would like to thank the reviewer for the nice comments.

We added section for the very limited available clinical trials.

Reviewer 4 Report

Comments and Suggestions for Authors

1. Delta9-THC is full agonist at CB1 receptors, while delta8-THC is partial agonist, therefore it can be similar activity as it is stated in the sentence (line 101-102). 

2. The Authors demonstrate only preclinical data using delta8-THC. Are there any clinical data regarding delta8-THC? If yes, please divide the pharmacology results into huan and animal/in vitro

3. CB1 and CB2 receptors and the effects induced by their activation should be more described.

4. Please provide some information on eventual treatment of delta8-THC-induced withdrawal

5. Please provide the limitations for the study.

Comments on the Quality of English Language

moderate correction is required

Author Response

  1. Delta9-THC is full agonist at CB1 receptors, while delta8-THC is partial agonist, therefore it can be similar activity as it is stated in the sentence (line 101-102). 

Observed.

  1. The Authors demonstrate only preclinical data using delta8-THC. Are there any clinical data regarding delta8-THC? If yes, please divide the pharmacology results into human and animal/in vitro.

We added section for the very limited available clinical trials.

  1. CB1 and CB2 receptors and the effects induced by their activation should be more described.

More detailed were added as directed.

  1. Please provide some information on eventual treatment of delta8-THC-induced withdrawal

We added section for the very limited available clinical trials.

  1. Please provide the limitations for the study.

New section for limitation was added as suggested.